# Oxidative Stress, Mitochondrial Quality Control, Autophagy, and Sirtuins in Heart Failure

**DOI:** 10.3390/ijms26199826

**Published:** 2025-10-09

**Authors:** Jan Krekora, Marcin Derwich, Jarosław Drożdż, Elzbieta Pawlowska, Janusz Blasiak

**Affiliations:** 12nd Department of Cardiology, Medical University of Lodz, 92-213 Lodz, Poland; ejkrekora@op.pl (J.K.); jaroslaw.drozdz@umed.lodz.pl (J.D.); 2Department of Pediatric Dentistry, Medical University of Lodz, 92-217 Lodz, Poland; marcin.derwich@umed.lodz.pl (M.D.); elzbieta.pawlowska@umed.lodz.pl (E.P.); 3Faculty of Medicine, Mazovian University in Plock, 09-240 Plock, Poland

**Keywords:** heart failure, sirtuins, autophagy, oxidative stress, mitochondria

## Abstract

Heart failure (HF) has become an emerging problem, especially in regions where life expectancy is increasing. Despite its prevalence, the mechanisms behind HF development are not well understood, which is reflected in the lack of curative therapies. Mitochondria, autophagy, and sirtuins form a crucial triad involved in HF pathogenesis, interconnected by oxidative stress. Identifying a common pathway involving these three components could be valuable in developing new treatment strategies. Since HF is the end result of several cardiovascular diseases, this review highlights the main HF precursors and explores the roles of mitochondrial quality control (mtQC), autophagy, and sirtuins in HF development. Dysfunctional mitochondria may play a key role by enhancing oxidative stress and influencing autophagy and sirtuins, both of which possess antioxidant properties. The dual nature of autophagy—its pro-life and pro-death roles—may contribute to different outcomes in HF related to oxidative stress. As mtQC, autophagy, and sirtuins may interact, we present data on their mutual dependencies in HF. However, the specificity of these interactions remains unclear and needs further investigation, which could help identify new therapeutic targets. In conclusion, the interplay between mtQC, autophagy, and sirtuins may be crucial in HF pathogenesis and could be leveraged in developing HF treatments.

## 1. Introduction

Heart failure (HF) is a growing medical and social issue worldwide, especially in areas where life expectancy has increased significantly in recent years [1]. It was recognized as a new epidemic in 1997 [2]. Globally, approximately 64 million people are affected by HF, with about one million new cases annually in the US [3]. Currently, around 6.7 million adults aged 20 and older in the United States live with HF, and this number is projected to rise to 8.7 million by 2030, 10.3 million by 2040, and 11.4 million by 2050. The estimated lifetime risk is 24%. The incidence and prevalence of HF vary across different populations [4]. For instance, the incidence of HF in White populations has remained stable since the 1960s. However, the increase in hospitalizations reflects better survival after HF diagnosis, resulting in more individuals requiring repeated hospital stays [5,6].

Heart failure is often the final stage of most heart syndromes. Therefore, known risk factors for heart diseases—including hypertension, diabetes, lack of physical activity, hyperlipidemia, and smoking—are linked to the development of HF [7]. Traditionally, HF is classified based on whether the left ventricular ejection fraction (LVEF) is preserved or reduced. HF with preserved LVEF (HFpEF, diastolic HF, LVEF ≥ 50%) is generally harder to manage than HF with reduced LVEF (HFrEF, systolic HF, LVEF ≤ 40%). Whether HFpEF should be considered a distinct entity within the EF spectrum remains a topic of debate [8]. Additionally, the term HF with mid-range EF (HFmrEF, LVEF 41–49%) was introduced by the American College of Cardiology, the American Heart Association, and the European Society of Cardiology [9]. Some other classifications of HF can also be used (reviewed in [10]). No matter the ejection fraction, HF can progress from a chronic, stable state—where symptoms are managed with treatment—to an acute, decompensated state, where symptoms suddenly worsen and urgent medical care is often required [11].

Although the exact mechanism of HF development is not fully understood, it involves a complex interplay of structural, functional, and molecular changes that gradually reduce the heart’s ability to pump blood effectively (reviewed in [10,12]). The most important aspects of HF pathogenesis include ventricular remodeling, hemodynamic overload, neurohormonal activation, inflammation, oxidative stress, metabolic disturbances, endothelial and microvascular dysfunction, along with some comorbidities such as hypertension, coronary artery disease, diabetes, and obesity. Therefore, HF pathogenesis involves multiple factors, as outlined in several recent reviews, e.g., [13]. However, this list of the most important aspects of HF pathogenesis suggests that molecular mechanisms may play a role in the overall development of HF in many ways.

According to recent global and regional guidelines, HF therapy should include a combination of pharmacological, non-pharmacological, and device-based strategies tailored to individual patient conditions and comorbidities. Guideline-Directed Medical Therapy (GDMT) for HFrEF involves angiotensin receptor-neprilysin inhibitors, beta-blockers, mineralocorticoid receptor antagonists, sodium-glucose cotransporter 2 (SGLT2) inhibitors (also used for HFpEF), and diuretics [14]. Besides SGLT2 inhibitors, medications for blood pressure control and managing comorbidities, especially atrial fibrillation, diabetes, and obesity, are used in HFrEF [15]. Non-pharmacological HF management involves lifestyle changes to prevent heart diseases and patient education for early detection of decompensation [16]. Device therapy for HF includes implantable cardioverter-defibrillators, cardiac resynchronization therapy, and remote monitoring devices [17]. Advanced therapies for HF include mechanical circulatory support, heart transplantation, and gene therapy [18,19]. However, all these therapies are either in early stages or not curative, despite being foundational. Worsening heart failure is a clear indication that new therapeutic targets and strategies are highly needed. The challenges in HF therapy mainly result from incomplete knowledge of the mechanisms of HF pathogenesis, which are driven by molecular events.

Oxidative stress is involved in the development of many diseases, but the direct cause-and-effect relationship between stress and disease, as well as the pathways through which oxidative stress impacts physiological processes, are often unclear. In many cases, mitochondria are closely linked to oxidative stress. Conversely, dysfunctional mitochondria are reported to occur in HF [20]. Sirtuins (SIRTs), a class of histone deacetylases located in the nucleus/nucleolus, mitochondria and cytosol, are involved in mitochondrial quality control and antioxidant defense. They are reported to play a role in HF pathogenesis [20]. The products of oxidative stress, reactive oxygen and nitrogen species (RONS), damage cellular components, and autophagy is a main mechanism responsible for removing these damaged and often toxic cellular components [21]. Therefore, similar to oxidative stress, autophagy is involved in the development of many diseases, including HF [22]. Therefore, mitochondria, sirtuins, and autophagy form a crucial triad in HF pathogenesis, and identifying pathways shared by these three components in HF may lead to the development of new therapeutic targets.

To prevent HF, it is crucial to identify pre-HF states and prodromal stages of the disease, which may also occur during its preclinical phase. Several precursors to HF have been identified (Figure 1). However, not all of these precursors are specific to HF, so further evaluation is necessary to assess HF risk. Additionally, since HF often results from other cardiovascular diseases, a prior diagnosis of these conditions increases the specificity of HF precursors. Clinically, cardiac hypertrophy is the most commonly used precursor for HF.

In this review of hypotheses, we highlight molecular mechanisms that may be involved in HF development, focusing on oxidative stress, mitochondrial homeostasis, and autophagy. Sirtuins are identified as key players in both mitochondria and autophagy. Common pathways among these three components are recognized and proposed as new potential targets for HF prevention and treatment. Our search strategy concentrated on mechanisms, clinical implications, biomarkers, and therapeutic approaches in heart failure, emphasizing oxidative stress, autophagy, mitochondrial homeostasis, and sirtuins. It was based on publications from PubMed, Embase, Google Scholar, ScienceDirect, and Cochrane Library. We used search strings including “heart failure” and one of the following terms: “oxidative stress,” “ROS,” “autophagy,” “mitochondria,” or “sirtuin.” Publications from the past 10 years were prioritized unless foundational studies were necessary. The species included were humans and relevant animal models. All article types, including original research, reviews, and meta-analyses, were considered, with no language restrictions.

## 2. Energy Metabolism, Oxidative Stress and Mitochondrial Dysfunction in Heart Failure

### 2.1. Energy Metabolism

Under normal conditions, the heart uses energy mainly generated from fatty acids oxidation (FAO) in mitochondria, while glucose constitutes 20–30% of energy substrates (Figure 2) [23].

Under normal conditions, glycolysis plays a minor role in energy production in the heart, but in certain situations like ischemia, physical overload, HF, or stress, it becomes more significant [24]. The heart generally exhibits metabolic flexibility that allows it to adapt to fuel availability by shifting its substrate preference [25]. It has been suggested that HF is associated with a metabolic shift in the heart from mainly using fatty acids to relying more on glucose for energy production [26]. This shift toward glycolysis in HF is a key aspect of metabolic remodeling in the heart and is linked to dysfunctional mitochondria [27]. However, the relationship between fatty acids and glucose utilization in HF shows conflicting results, which is expected given the variety of HF clinical presentations. Recent studies indicate that these discrepancies are mainly due to methodological differences, and direct measurements consistently show active or increased FAO in HFpEF hearts, while glucose oxidation becomes impaired [28]. Therefore, the balance between fatty acids and glucose utilization, as well as between oxidative phosphorylation (OXPHOS) and glycolysis, may be critically connected to the development and severity of HF [29]. Notably, ATP levels in a failing heart are maintained until the final stages, despite early signs of a mismatch between energy supply and demand [30,31]. Additionally, ketone bodies can serve as energy sources in both healthy and failing hearts [32]. In all these processes, mitochondria are essential.

### 2.2. Oxidative Stress

Impaired mitochondria are directly linked to oxidative stress, a state where RONS production exceeds their neutralization. This may result from external factors such as radiation and certain chemicals, or be induced internally by disturbances in metabolic processes, especially those related to energy production. Excessive RONS can damage cellular molecules, including proteins, lipids, and nucleic acids. To protect against oxidative stress, cells have developed an antioxidant system with three main components: antioxidant enzymes, DNA repair proteins, and low-molecular-weight antioxidants, which collectively keep RONS levels normal. Dysfunction in any of these elements can cause pathological events. Cells constantly produce RONS as byproducts of many physiological processes, and mitochondrial complexes generate RONS during their normal operation. However, damage to the complexes in the mitochondrial electron transport chain (ETC) can cause RONS overproduction, potentially damaging ETC proteins and their encoding genes. This can create a “vicious cycle,” leading to cell death [33]. During normal function, complexes I and III of the ETC leak electrons, which can produce mitochondrial superoxide [34]. This superoxide is converted by superoxide dismutase 2 (SOD2) into hydrogen peroxide, another RONS, which is then neutralized by peroxiredoxin (PRX) and glutathione peroxidase (GPX) [35]. Damage caused by mitochondrial RONS is considered a key mechanism in HF development, as shown in several studies on failing hearts in HF patients and animal models [20]. RONS scavenging in mitochondria has been demonstrated to benefit HF animal models [36,37,38]. Reduced nicotinamide adenine dinucleotide (NADH) plays an important role in the functioning of the PRX and GPX systems in mitochondria, and its supply requires two enzymes: isocitrate dehydrogenase 2 (IDH2) and nicotinamide nucleotide transhydrogenase (NNT). Impairment of these has been linked to HF development [39]. However, increased NNT activation in the mouse failing heart under oxidative stress has also been observed [40]. This may disconnect NADPH from ATP production pathways and impair energy metabolism in the HF heart during oxidative stress. Under normal conditions, NNT transfers reducing equivalents from NADH to NADP^+^, generating NADPH in a reaction coupled to the mitochondrial proton gradient, supporting antioxidative defense without compromising ATP production. In pathological states, such as pressure overload-induced heart failure, the metabolic demand and mitochondrial stress cause NNT to consume instead of produce NADPH to regenerate NADH, which is used to stimulate ETC for ATP synthesis at the expense of NADPH.

A failing heart needs more energy, and stimulating mitochondrial ETC is linked to increased oxidative stress due to more RONS leaking from overactive ETC complexes. However, because of increased protein acetylation, ATP synthase is inhibited in HF [41]. This leads not only to reduced energy production but also to more RONS generated because of impaired electron flow in the ETC. Animal studies showed that the mitochondrial antioxidant system, including SOD2, was compromised in HF [36]. Human studies show decreased SOD2 activity in HF or no change [42,43]. Additionally, studies on myocardial tissue homogenates from the left ventricular wall of hearts with end-stage failure caused by dilated or ischemic cardiomyopathy showed increased levels of catalase, the main enzyme responsible for breaking down hydrogen peroxide, at both the mRNA and protein levels [44].

In summary, several studies report the involvement of oxidative stress in HF pathogenesis, and ample evidence suggests that stress may be both a cause and a consequence of the disease. However, the exact mechanisms behind this involvement remain unclear, as does its origin in HF. Consistent reports indicate that impaired mitochondria may be a primary source of RONS in HF. Human and animal studies suggest that hyperacetylation of proteins essential for mitochondrial homeostasis may contribute to oxidative stress in HF. Additionally, Ca^2+^ overload and defective removal of damaged mitochondria through mitophagy can increase RONS levels. If mitochondrial damage is not repaired, it may lead to more severe damage in a “vicious cycle” or “RONS-induced RONS release.” These issues are discussed in more detail in the next section.

### 2.3. Mitochondrial Dysfunction

Oxidative stress can be both a cause and a result of mitochondrial impairment. Therefore, maintaining mitochondrial homeostasis is crucial for HF development. This is achieved through mitochondrial quality control (mtQC), a comprehensive system of cellular processes that protect mitochondrial integrity, function, and adaptability based on physiological needs and stress [45]. This system ensures mitochondria’s effectiveness in energy production while minimizing damage from reactive oxygen species and other stress factors. The main components of mtQC are mitochondrial biogenesis, mitochondrial dynamics (fusion and fission), and mitophagy. Additionally, the maintenance of mitochondrial DNA (mtDNA), mitochondrial proteostasis, and unfolded protein response also play a role in mtQC [46]. Several studies report mtQC impairments in HF.

Peroxisome proliferator-activated receptor gamma coactivator 1-alpha (PGC-1α) is a key regulator of mtQC [47]. PGC-1α is highly expressed in the heart, promoting fatty acid oxidation and mitochondrial respiration [48]. Although PGC-1α lacks intrinsic enzymatic activity and a DNA-binding domain, it plays a critical role in mitochondrial biogenesis through its interaction with transcription factors nuclear respiratory factor 1 and 2 (NRF1/NRF2) and mitochondrial transcription factor A (TFAM) [49]. Additionally, PGC-1α is essential for mitochondrial dynamics and mitophagy.

Studies on HF patients and animal models have shown inconsistent results, with some indicating decreases and others showing no effect (reviewed in [50]). However, a 2018 study found that PGC-1α levels after transverse aortic constriction (TAC), a primary animal model for cardiac hypertrophy and heart failure, fluctuated over 5–14 days during HF progression [50]. This fluctuation may partly explain discrepancies between findings across different studies and suggests that research on PGC-1α in HF should focus on its time dependence rather than on “static” measurements. Animal studies revealed that mice with a double PGC-1α knockout experienced more severe cardiac dysfunction and higher mortality rates under stress conditions, such as TAC, compared to normal mice [51,52]. Conversely, overexpression of PGC-1α appeared to promote the development of HF [53,54]. Other potential reasons for these discrepancies include the complexity of regulation—expression levels do not always match activity—different stages of disease may variably influence PGC-1α expression, metabolic and inflammatory signals might cause bidirectional modulation, and variations in experimental design can lead to measurement inconsistencies.

PTEN-induced kinase 1 (PINK1) is an important factor in mtQC because it initiates mitophagy, a process for the selective removal of damaged mitochondria, and plays a key role in the development of cardiovascular disease [55]. It was observed that PINK1 was entirely localized to the mitochondria, and its levels decreased in the end-stage of HF in humans [56]. PINK1^−/−^ mice showed increased oxidative stress, fibrosis, cardiomyocyte apoptosis, mitochondrial dysfunction, and developed left ventricular failure, along with signs of cardiac hypertrophy by 2 months of age. These studies clearly demonstrate that PINK1 activity is essential for postnatal myocardial development, maintaining mitochondrial function, and redox balance in cardiomyocytes. Mitophagy was reported to be impaired in HF in several other studies (reviewed in [57]). This problem will be discussed in more detail later, focusing on the role of autophagy in HF.

The mitochondrial DNA is a double-stranded, short (16,569 base pairs), closed DNA molecule that occurs in several copies within the mitochondria. Although the proximity of the electron transport chain (ETC) makes mtDNA more exposed to external DNA-damaging agents than nuclear DNA, its maintenance is less efficient. The DNA damage response (DDR), the main pathway for DNA maintenance in mitochondria, is not fully understood but involves fewer subpathways and proteins than the nuclear DDR. A reduced number of mtDNA copies was observed in a mouse model of myocardial infarction and remodeling created by ligation of the left anterior descending coronary artery [58]. Also, there is a decrease in the transcripts of mtDNA-encoded genes, including subunits of complex I, complex III (cytochrome b), and 12S and 16S rRNA. These changes and related adverse phenotypic alterations are attributed to increased oxidative stress associated with HF as an end-stage of myocardial infarction. N-glycosylase/DNA lyase (8-oxo-deoxyguanosine glycosylase, OGG1) is the primary mammalian enzyme responsible for repairing oxidative DNA damage. It has 8 isoforms, with 7 targeting mitochondria, and the OGG1-2a isoform being the most abundant. An increased migration of OGG1-2a to mitochondria was observed in the early stage of compensated cardiac hypertrophy induced by abdominal aortic constriction [59]. This helped maintain mtDNA integrity despite increased oxidative stress. Therefore, increased migration of OGG1-2a may be seen as a mechanism that supports cardiac function during the compensatory stage. A direct causal link between mtDNA damage and heart failure (HF) was shown in a study where transgenic mice with Tet-on inducible, cardiomyocyte-specific expression of a mutant uracil-DNA glycosylase 1 (mutUNG1) were created [60]. In addition to uracil, this mutated UNG1 variant also removes thymine from mtDNA, leading to transitional apyrimidinic sites that impair mtDNA functions. After inducing mutUNG1 in cardiac myocytes, the mice developed hypertrophic cardiomyopathy, which progressed to congestive heart failure and resulted in premature death within about two months. The affected hearts showed reduced mtDNA replication and transcription, suppressed mitochondrial respiration despite higher levels of PGC-1α, mitochondrial mass, and antioxidant enzymes, and impaired mitochondrial fission and fusion dynamics. These changes caused worsening myocardial contractility, serving as the mechanism behind heart failure.

Increased mitochondrial damage and impaired mtQC, leading to the release of mtDNA into circulation, may contribute to cardiac inflammation, as it can trigger activation of circulating immune cells [61]. As inducers of the NLRP3 inflammasome disrupted normal mitochondrial homeostasis, which decreased the levels of the coenzyme NAD^+^ and inactivated the NAD^+^-dependent SIRT2 [62]. This effect caused the buildup of acetylated α-tubulin, which helped dynein-dependent mitochondrial transport and the positioning of NLRP3 adaptor on mitochondria relative to NLRP3 on the endoplasmic reticulum. Since HF is linked with NAD^+^/NADH redox imbalance, overactivation of NLRP3 might be another factor, besides mtDNA release, that connects mitochondrial dysfunction with inflammation in HF.

The involvement of mitochondria in HF pathogenesis goes beyond mtQC, in general, and PGC-1α specifically. The acetylation of mitochondrial proteins, mainly at lysine residues and primarily facilitated by acyl-CoA molecules, is crucial not only for mtQC but also for enzyme activity, including SOD2, metabolic adaptation, and aging [63]. Recently, it was shown that the levels of sirtuin 3 (SIRT3), a key regulator that removes acetyl groups from lysine residues, and nicotinamide adenine dinucleotide (NAD^+^) decreased in HF in dogs, contributing to a hyperacetylation state [64]. Hyperacetylation of mitochondrial proteins was also observed in other studies involving HF patients and animal models [65,66,67]. The mechanism behind the increased acetylation of mitochondrial proteins in HF is not entirely understood, but it may be due to increased levels of acyl-CoAs and/or heightened activity of SIRT3 in the failing heart [20]. The activity of each sirtuin depends on NAD^+^ availability, and decreased NAD^+^ levels along with a reduced NAD^+^/NADH ratio have been observed in hearts with mitochondrial dysfunction [68]. The NAD^+^/NADH redox imbalance promotes activation of the NLRP3 inflammasome, contributing to a vicious cycle and shifting mitochondria from energy producers to initiators of cell death [20].

Another mitochondria-related hallmark of HF is the deregulation of Ca^2+^ homeostasis [69]. Impaired Ca^2+^ reuptake by the sarcoplasmic reticulum (SR) in a failing heart results in decreased cytosolic Ca^2+^ transients. Since mitochondria are located near the SR, it is believed that, under pathological conditions, the SR might act as a Ca^2+^ sink. This Ca^2+^ overload then leads to mitochondrial dysfunction [20]. As the Ca^2+^ ions decrease, PHPPP declines. Reduced mitophagy regulates many mitochondrial enzymes and other proteins, including those involved in oxidative defense, OXPHOS, cardiomyocyte death, and the mitochondrial permeability transition pore (mtPTP) opening, which are important in HF pathogenesis [70]. The mitochondrial Ca^2+^ uniporter (MCU) and mitochondrial Na^+^/Ca^2+^ exchanger (NCLX) are two main components involved in Ca^2+^ transport within mitochondria [71]. However, studies on the involvement of both transporters in HF pathogenesis have yielded inconsistent results, and overall, the mechanism of how Ca^2+^ overload causes damage in mitochondria remains unclear and requires further investigation (reviewed in [20]).

Mitochondria interact with other organelles such as the endoplasmic reticulum, lysosomes, ribosomes, lipid droplets, and the nucleus, and this interaction helps maintain cardiac homeostasis. Its dysfunction may play a role in HF development (reviewed in [72]).

In summary, mitochondrial dysfunction in HF hearts, including defective mtQC, may lead to oxidative stress, increased mtDNA damage, NLRP-mediated inflammation, calcium overload, disrupted NAD+/NADH redox balance, impaired fusion and fission processes, and reduced mitophagy (Figure 3). It is important to emphasize that the cause-and-effect relationships among these effects related to dysfunctional mitochondria, oxidative stress, and HF are not entirely clear and require further investigation.

Despite a clear link between HF pathogenesis and mitochondrial dysfunction, there is no established therapeutic approach targeting mitochondrial problems in HF. However, Clinical.Trials.gov lists some ongoing or completed trials aimed at restoring mitochondrial health, enhancing energy production, and reducing oxidative stress in cardiomyocytes in HF (https://clinicaltrials.gov/search?cond=Heart%20Failure&term=mitochondria&intr=treatment, accessed on 30 August 2025).

## 3. Sirtuins in Heart Failure

Sirtuins (silent information regulators, SIRTs) are a family of NAD+-dependent class III histone deacetylases and adenosine diphosphate ribosyl transferases, consisting of seven members, SIRT1-7 in mammals [73]. They are located in the nucleus, nucleolus, cytoplasm, and mitochondria, with most exhibiting dual localizations. They perform numerous functions, but their main role is to control gene expression through the deacetylation of chromatin proteins and transcription factors (reviewed in [74]). Other functions relate to their roles in antioxidant defense, DDR, autophagy, apoptosis, inflammation, metabolism, aging, and brain function, although these are largely driven by their gene expression regulation activities. Beyond physiological roles, sirtuins are also involved in the development of many disorders, including cancer and neurodegenerative diseases [75,76]. However, in both of these syndromes and other pathologies, sirtuins can have either beneficial or adverse effects.

Some works cited in this section relate directly to syndromes that are precursors or associates of HF, including cardiac hypertrophy, ischemia–reperfusion damage, and fibrosis. It is justified that in most, if not all, cases, HF is linked to at least one of these symptoms.

Sirtuins are essential in heart development, primarily by controlling the expression of genes vital for this process. For example, SIRT1 deficiency has been linked to congenital heart defects [77]. These proteins are involved from the very start of heart formation, as they participate in the differentiation of cardiac progenitor cells [78]. Additionally, sirtuins influence heart morphogenesis by regulating key signaling pathways such as wingless-type MMTV integration site family (WNT), neurogenic locus notch homolog protein (NOTCH), and forkhead box protein O (FOXO) [79,80]. Their antioxidant properties help protect the developing heart from oxidative stress [81]. Sirtuins also manage energy metabolism during heart development, especially by promoting mitochondrial biogenesis and fatty acid oxidation (FAO), which are crucial for meeting the energy demands of the growing heart [82]. These functions of sirtuins will be explored further later, focusing on mitochondrial activity and autophagy.

Generally, SIRTs are considered to play a beneficial and important role in suppressing HFpEF, but notable exceptions to this rule have been observed [83]. Exercise training and caloric restriction have been shown to improve the quality of life in patients with HFpEF [84]. Two clinical trials demonstrating that a well-structured 4-week exercise training–caloric restriction program led to reductions in systolic blood pressure and increased EF suggest that these effects were partly due to higher NAD+ levels and greater SIRT1 activity [85,86]. Additionally, an increase in antioxidant capacity was observed in one of these studies. SIRT1 was shown to protect against the harmful effects caused by cardiac remodeling in a mouse model of HFpEF by regulating lipid metabolism and inflammation [87].

There are issues with clearly classifying SIRT2 as either a protective or harmful factor for cardiac function [88]. Also in HF, the activity of SIRT2 was reported to have both beneficial and adverse effects. It was demonstrated that the hearts of the SIRT2^−/−^ mice showed improved cardiac function after ischemia–reperfusion (I/R) and pressure overload (PO), suggesting that SIRT2 had maladaptive effects in the heart in response to stress [89]. Similar outcomes were observed in mice lacking SIRT2 specifically in cardiomyocytes. Further research showed that SIRT2 affects the levels and activity of NRF2 within cells, leading to a reduction in antioxidant enzyme expression. Deleting NRF2 from the hearts of SIRT2^−/−^ mice eliminated their protection after PO. Additionally, giving a SIRT2 inhibitor to mouse hearts decreased cardiac size and lowered hypertrophy caused by PO. Therefore, SIRT2 seems to have harmful effects on the heart and influences its response to injury and the development of cardiac hypertrophy, which is a step toward HF. Conversely, SIRT2 was downregulated in hypertrophic hearts from mice [90]. SIRT2^−/−^ mice showed cardiac hypertrophy and fibrosis, along with decreased cardiac ejection fraction and fractional shortening. Conversely, overexpressing SIRT2 specifically in the heart protected against hypertrophy and fibrosis caused by age and angiotensin II (ANGII), while also restoring cardiac function. Mechanistically, SIRT2 helps maintain AMP-activated protein kinase (AMPK) activity in hypertrophic hearts both in vivo, whether due to aging or ANGII, and in cardiomyocytes in vitro. The liver kinase B1 (LKB1), a key kinase upstream of AMPK, was identified as a direct target of SIRT2. Consequently, SIRT2 promotes AMPK activation by deacetylating LKB1. Loss of SIRT2 reduces AMPK activity, increasing susceptibility to aging-related and ANGII-induced cardiac hypertrophy. Elevated plasma SIRT2 levels positively correlate with heart failure (HF) occurrence in patients after an acute myocardial infarction [91]. A lower level of SIRT2 was observed during pathological heart hypertrophy in mice [92]. Animals lacking SIRT2 developed natural heart issues with hypertrophy, remodeling, fibrosis, and dysfunction as they aged. Young SIRT2-deficient mice also showed more severe hypertrophy when exposed to stimuli. Conversely, increasing SIRT2 levels decreased agonist-induced cardiac hypertrophy in cardiomyocytes through a cell-intrinsic process. Mechanistically, SIRT2 interacted with and deacetylated the NFATc2 transcription factor. The absence of SIRT2 stabilized NFATc2 and increased its presence in the nucleus, boosting its transcription activity. Therefore, inhibiting NFAT can restore heart functions in SIRT2-deficient mice.

The downregulation of SIRT3 was shown to promote fibrotic effects in a mouse model of HFpEF, linked to hyperacetylation of mitochondrial proteins, which leads to increased production of interleukins IL1B and IL18 and enhanced assembly of NLRP3 on hyperacetylated mitochondria [93]. Elevated β-hydroxybutyrate levels decreased NLPR3 formation and triggered proinflammatory cytokine-driven mitochondrial dysfunction and fibrosis. Additionally, β-hydroxybutyrate lowered the acetyl-CoA pool and mitochondrial acetylation, partly through activating citrate synthase and inhibiting fatty acid uptake. A disruption in mitochondrial fatty acid oxidation associated with hyperacetylation of key enzymes in this pathway was shown in a mouse HFpEF model [94]. Downregulation of SIRT3 and a shortage of NAD+ contribute to the observed hyperacetylation of mitochondrial proteins. Reduced expression of genes involved in NAD+ synthesis was confirmed in cardiac tissue from HFpEF patients. HFpEF mice supplemented with nicotinamide riboside, a NAD+ precursor or an activator of NAD+ synthesis, showed improvements in mitochondrial functions and a reversal of the HFpEF phenotype. Therefore, HFpEF is linked to mitochondrial dysfunction in the myocardium and NAD+ deficiency, which may be related to SIRT3 downregulation. It was demonstrated that supplementation with nitrite and metformin, which activate the skeletal muscle SIRT3-5′-AMPK pathway, improved glucose uptake and metabolism in a mouse model of HFpEF [95]. In similar research, SIRT3 deficiency in skeletal muscle increased the secretion of lysyl oxidase homolog 2 and β2-microglobulin, contributing to pulmonary vascular remodeling in pulmonary hypertension caused by HFpEF [95]. A decrease in indole-3-propionic acid, which activates the nicotinamide N-methyl transferase (NNMT)-SIRT3 axis, was observed in HFpEF patients [96]. These studies were further confirmed by nicotinamide supplementation, *NNMT* knockdown and overexpression in a mouse model of HFpEF.

SIRT4 is primarily located in mitochondria but can also be found in the nucleus, cytoplasm, centrosomes, and microtubules [97]. Although SIRT4 is highly expressed in the heart, its roles in cardiac development and disease are not yet fully understood. SIRT4 may influence cardiac processes such as ischemia–reperfusion and remodeling differently than other sirtuins [98]. It has been shown that SIRT4 overexpression in mice undergoing angiotensin II infusion promoted the progression from compensated to decompensated cardiac hypertrophy [99]. Recently, it was demonstrated that specifically overexpressing SIRT4 in cardiomyocytes accelerated the development of heart failure in response to pressure overload, mainly through a mitochondrial RONS-mediated increase in profibrotic transcriptional signaling [100]. These findings suggest that SIRT4 could be a therapeutic target whose inhibition might prevent the development or progression of HF in individuals with cardiac SIRT4 overexpression. Since overexpressing SIRT4 in cardiomyocytes speeds up HF, especially under pressure overload, it might seem that increasing NAD+ levels could worsen SIRT4-related HF because SIRT4 depends on NAD+. However, NAD+ supplements improve mitochondrial function and ATP production, activate SIRT1 and SIRT3 to boost antioxidant defenses and mitochondrial autophagy, and reduce cardiac dysfunction and fibrosis in various HF models, including pressure overload and ischemic cardiomyopathy. Therefore, SIRT4 overexpression—rather than its activation by NAD+—mainly drives HF pathology. NAD+ supplementation does not specifically activate SIRT4 but instead supports a broader network of NAD+-dependent enzymes, many of which are protective of the heart.

Fibrosis in heart failure (HF) was studied in another research involving SIRT5 [101]. In this study, a mouse strain overexpressing SIRT5 (SIRT5OE) was subjected to chronic pressure overload through TAC. Compared to controls, SIRT5OE mice were protected against TAC effects, such as left ventricular dilation and reduced ejection fraction. Transcriptomic analysis revealed that SIRT5 suppresses key HF-related consequences: the metabolic shift from fatty acid oxidation to glycolysis, immune activation, and fibrotic signaling pathways. Oxidative stress in mouse cardiomyocytes caused SIRT5 downregulation [102]. SIRT5 knockdown decreased cell viability and increased apoptotic cells and caspase 3/7 activity. A direct interaction between Bcl-2-like protein 1 (B2CL1) and SIRT5 was observed. Therefore, SIRT5 may negatively regulate oxidative stress-induced apoptosis in cardiomyocytes and could be involved in preventive and therapeutic strategies for oxidative stress-related cardiac injury, such as in HF. Ischemia–reperfusion injury (IRI) often precedes HI. A cardioprotective role of mitochondrial SIRT5 in a mouse model of cardiac IR injury has been reported [103]. SIRT5 downregulation in IR-injured hearts was associated with increased protein lactylation and mitochondrial dysfunction. Conversely, raising SIRT5 levels reduced mitochondrial damage and alleviated cardiac injury. SIRT5 interacts with the mitochondrial protein adenine nucleotide translocase 2 (ANT2), inhibiting its lactylation and strengthening its interaction with another mitochondrial protein, voltage-dependent anion-channel 1 (VDAC1). Lactylation-resistant ANT2 forms a more effective complex with VDAC1, improving cardiac function after injury. In conclusion, SIRT5 promotes the interaction between mitochondrial proteins ANT2 and VDAC1 to preserve mitochondrial homeostasis and support heart function in IRI.

Endothelial SIRT6 plays a role in transporting fatty acids across the endothelial barrier by suppressing peroxisome proliferator-activated receptor gamma (PPARγ, encoded by the PPARG gene) [104]. This mechanism is crucial for mitigating experimental HFpEF by restoring endothelial SIRT6 function, as shown by the deacetylation of histone H3K9 around the PPARG promoter. Similar research involving diabetic HF patients observed a decrease in SIRT6 in cardiac tissues [105]. Studies also found that restoring endothelial SIRT6 expression in a mouse model of HFpEF and diabetes—induced by a long-term high-fat diet combined with low-dose streptozocin—improved diastolic function and reduced cardiac lipid buildup. SIRT6 lowered endothelial fatty acid uptake by suppressing endothelial PPARγ expression through SIRT6-dependent deacetylation of histone H3 near the PPARγ gene promoter. This suggests that decreased endothelial SIRT6 links diabetes to HFpEF by affecting fatty acid transport across the endothelial barrier via PPARγ. Additionally, SIRT6, telomerase reverse transcriptase (TERT), and telomere repeat binding factor 1 (TRF1) were decreased in TAC mice [106]. Overexpressing SIRT6 increased TERT and TRF1 levels and improved survival after TAC. Further analysis showed that SIRT6 overexpression reduced TAC-induced heart problems and inflammation, which led to less scarring and smaller infarcts.

SIRT7 has this unique feature among other SIRTs: it can be localized in the nucleolus [107]. Although there is no direct evidence of the specific role that this unique localization of SIRT7 may have in cardiac function, it cannot be ruled out, as it is linked to protein synthesis, stress responses, and chromatin dynamics [108]. An increased expression of SIRT7 was observed in cardiomyocyte-specific SIRT7-knockout mice subjected to pressure overload induced by TAC [109]. These animals also showed an increase in heart weight relative to tibial length and exhibited reduced cardiac contractile function compared to controls. A direct interaction between SIRT7 and the transcription factor GATA-4 was identified, and *GATA4* knockdown lessened the severity of phenylephrine-induced cardiac hypertrophy. SIRT7 deacetylated GATA4 in cardiomyocytes, affecting its transcriptional activity. This suggests that SIRT7 deficiency in cardiomyocytes promotes hypertrophy under pressure overload, and SIRT7’s anti-hypertrophic effects are mediated through its interaction with and promotion of GATA4 deacetylation. In another study, SIRT7-deficient animals showed a reduction in lifespan and developed heart hypertrophy and inflammatory cardiomyopathy [110]. Hearts of SIRT7 mutants showed widespread fibrosis. The study also demonstrated that SIRT7 interacts with the cellular tumor antigen p53 and deacetylates it in vitro, resulting in hyperacetylation of p53 in vivo and an increased rate of apoptosis in the mutant mice’s myocardium. Primary cardiomyocytes lacking SIRT7 exhibited higher baseline apoptosis and were less resistant to oxidative and genotoxic stress, indicating that SIRT7 is crucial for regulating stress responses and cell death in the heart.

Major effects of SIRTs related to HF are summarized in Table 1. The selection of these effects presented in this section is subjective and not exhaustive. More information can be found elsewhere, e.g., [74,81,111]. In summary, SIRTs are important players in HF pathogenesis, and most studies suggest their beneficial potential and protective effects against HF. However, due to incomplete knowledge of the basic mechanisms of SIRTs homeostasis, further research is necessary to determine their potential in HF, especially regarding their mutual interactions.

## 4. Autophagy in Heart Failure

Autophagy is a process in which cells remove damaged, dysfunctional, or unnecessary material, as well as invaders and their remnants, through a central molecular pathway essential for maintaining cellular and organismal homeostasis [112]. Autophagy occurs as either degradative autophagy or secretory autophagy. In degradative autophagy, materials to be removed are broken down inside the cell and may then be recycled. In secretory autophagy, the cargo is exported out of the cell. Research on secretory autophagy is still in the early stages of understanding its basic mechanisms, and its involvement in HF has not been definitively confirmed [113]. Therefore, we will concentrate exclusively on degradative autophagy.

Degradative autophagy is categorized into three types: macroautophagy (commonly called autophagy), microautophagy, and chaperone-mediated autophagy (CMA) (Figure 4). The primary components of the autophagy machinery include sequestosome 1 (SQSTM1/p62), optineurin (OPTN), ubiquilin 2, nibrin 1, WD repeat and FYVE domain-containing 3, calcium-binding and coiled-coil domain 2, mechanistic target of rapamycin (MTOR), and huntingtin [112]. A key feature of degradative autophagy is the formation of autolysosomes through the fusion of autophagosomes with lysosomes, a process that involves SNARE proteins such as syntaxin 17, SNAP29, and vesicle-associated membrane proteins 7 and 8 [114]. This system directs cargo through specific receptors or adaptors that recognize degradation signals on cargo proteins and bind to LC3 and γ-aminobutyric acid receptor-associated protein on autophagosomes [115].

Autophagy is closely connected to regulating heart structure and function in various physiological and pathological conditions, including heart failure and its precursors (reviewed in [116]). The importance of autophagy in HF development is especially notable due to the increasing prevalence of HF and the decline of autophagy with age [117]. However, our understanding of how impaired autophagic flux leads to cardiac dysfunction remains limited. Typically, the harmful effects of disrupted autophagy in the heart are considered to involve the accumulation of damaged proteins and organelles that are toxic to cardiomyocytes, but it is unclear why and how this occurs in HF.

Rapamycin is used to stimulate autophagy, while chloroquine suppresses it [118]. It was observed that ANGII-infused mice showed increased co-localization of LC3 puncta with vimentin [119]. Furthermore, treating cardiac fibroblasts with rapamycin increased type I collagen (COLI) and decreased fibronectin (FN) after ANGII stimulation. Similarly, inhibiting autophagy with chloroquine or knocking down ATG5 worsened ANGII-induced buildup of COLI and FN. Additionally, rapamycin improved cardiac fibrosis and dysfunction in mice infused with ANGII, while chloroquine worsened both fibrosis and dysfunction and also impaired heart function. This suggests that autophagy may play a protective role in reducing extracellular matrix buildup in the heart. In similar studies, activating the AMPK/MTOR/ULK1 pathway to stimulate autophagy decreased collagen deposits and myocardial fibrosis in mice [120].

Hypertrophic cardiomyocytes, typical of HF and its prodrome, accumulate misfolded proteins and damaged organelles [121]. These potentially toxic cellular debris are removed by the combined action of the ubiquitin–proteasome system and autophagy. Therefore, impaired autophagy may be both a cause and a consequence of cardiac hypertrophy. Several hypertrophy-related signaling pathways overlap with those involved in autophagy regulation [122]. The Ca^2+^/calcineurin signaling pathway was shown to activate the transcription of hypertrophy-related genes and inhibit autophagy in cardiomyocytes, involving AMPK [123,124]. MTOR, a critical autophagy protein, is, along with glycogen synthase kinase-3 (GSK3), a downstream target that underpins the involvement of the PI3K/AKT pathway in cardiac hypertrophy [125]. Cardiac autophagy and mitophagy might be controlled by mitogen-activated protein kinases (MAPKs), which also play a key role in hypertrophic signaling [126,127]. Therefore, several signaling pathways connect cardiac hypertrophy with autophagy, but more research is needed to establish a clear cause-and-effect relationship between these processes. Deletion of the ATG5 gene in the mouse heart led to cardiac hypertrophy and functional problems, which resulted in HF and ultimately animal death [128]. This confirms that autophagy may decrease during cardiac hypertrophy. In the final stage of HF, the myocardium becomes acutely decompensated, which is linked to the overproduction of misfolded proteins, damaged organelles, and other RONS-related products [129]. This may trigger an excessive autophagic response, potentially damaging normal proteins and organelles, which can lead to apoptosis and loss of cardiomyocytes [130]. Therefore, autophagy may have both beneficial and harmful effects in HF, reflecting its pro-life and pro-death properties [131]. Autophagy supports cell survival and maintains homeostasis during stress or nutrient deprivation by recycling nutrients, regulating organelle quality, clearing protein aggregates, and assisting immune defense, development, and differentiation. It can also cause cell death, especially when dysregulated or excessively activated. The main harmful roles of autophagy include autophagic cell death (type II programmed cell death), tumor suppression, neurodegeneration, and ischemia/reperfusion injury. However, autophagy is not inherently good or bad; its effects depend on the level and duration of activation, cell type, physiological state, and the presence of other stress signals or death pathways.

A combined in vivo and in vitro study was conducted to examine the role of epigenetic modifications in cardiac fibrosis [132]. The involvement of miR-17-5p and BCL2/adenovirus E1B 19 kDa protein-interacting protein 3 (BNIP3) in modulating mitophagy and alleviating pathological cardiac fibrosis was studied in the TAC mice model of myocardial fibrosis and in cardiac fibroblasts treated with ANGII. Lower levels of myocardial miR-17-5p were associated with decreased left ventricular systolic function and increased collagen buildup in heart tissue. In vitro, treatment with angiotensin II led to reduced expression of miR-17-5p, increased BNIP3 levels, and excessive mitophagy, as shown by higher RONS levels, decreased ATP production, and increased markers of fibrosis. Additionally, overexpression of miR-17-5p directly bound to the 3′-untranslated region of BNIP3, significantly reducing its expression, which helped restore mitochondrial balance and decrease collagen synthesis. Conversely, overexpressing BNIP3 negated the anti-fibrotic and mitochondrial-protective effects of miR-17-5p. Therefore, the miR-17-5p/BNIP3 signaling pathway regulates mitophagy in cardiac fibroblasts and is essential for fibrotic remodeling. As a result, it may serve as a target for therapeutic strategies aimed at reducing cardiac fibrosis and delaying heart failure progression. In another epigenetic-related study, it was observed that the miRNA-212/132 family influenced cardiac hypertrophy and autophagy in cardiomyocytes [133]. MiR-212/132 null mice were resistant to pressure-overload-induced heart failure (HF), while cardiomyocyte-specific overexpression of the miR-212/132 family caused pathological cardiac hypertrophy, HF, and death. Both miR-212 and miR-132 directly suppress the anti-hypertrophic and pro-autophagic transcription factor FOXO3. Overexpressing these miRNAs led to excessive activation of the pro-hypertrophic calcineurin/NFAT signaling pathway and impaired autophagy during starvation. Consequently, inhibiting miR-132 with antagomir injections could potentially restore normal cardiac hypertrophy and reduce HF, offering a promising therapeutic strategy for this condition.

The importance of the FOXO3-BNIP3 pathway in mitochondria-related heart failure development was confirmed in another study showing a FOXO3-driven increase in BNIP3 in normal and phenylephrine (PE)-stressed cardiomyocytes from a rat HFpEF model [134]. This effect was linked to increased mitochondrial Ca^2+^, which led to reduced mitochondrial membrane potential, mitochondrial fragmentation, and cell death. A cardiotropic adeno-associated virus serotype 9 encoding dominant-negative FOXO3 (AAV9.dn-FX3) was used for gene delivery in the HFpEF model. While dn-FX3 lessened the rise in BNIP3 levels and its effects in PE-stressed ACM, delivering AAV9.dn-FX3a in an experimental HFpEF model decreased BNIP3 expression. This treatment reversed harmful left ventricular remodeling and improved both systolic and especially diastolic function, with additional benefits in mitochondrial structure and activity. Furthermore, FOXO3a promotes the expression of harmful genes related to mitochondrial apoptosis, autophagy, and cardiac atrophy. Therefore, FOXO3 may connect HF, mitochondria, and autophagy.

Oxidative stress triggers autophagy, and ANGII increases cardiac mitochondrial RONS production, leading to a decline in mitochondrial membrane potential in cardiomyocytes and resulting in greater oxidative damage to cardiac mitochondrial proteins and mtDNA deletions [135]. These harmful effects of ANGII on mitochondria were associated with an increase in the number of autophagosomes and amplified mitochondrial biogenesis, which may help replace damaged mitochondria and restore energy production. This study also highlighted the key role of RONS produced by mitochondria, as shown by experiments with mice overexpressing catalase targeted to mitochondria, compared to those overexpressing peroxisomal-targeted catalase. The mice with mitochondrial-targeted catalase, but not those with peroxisomal-targeted catalase, were resistant to cardiac hypertrophy, fibrosis, mitochondrial damage, and heart failure.

Ventricular remodeling plays a key role in the development and progression of HF [136]. It is associated with oxidative stress, ER stress, and a weakened ubiquitin–proteasome system, making autophagy a crucial process for mitigating the effects of remodeling. In ventricular remodeling, autophagy can be both beneficial and detrimental, acting as a protective mechanism in early stages but potentially causing cell death and heart failure if it becomes uncontrolled [137].

Impaired autophagy in mice caused by knocking out ATG3, a key autophagy initiator, led to NAD+ deficiency in the heart due to increased NAD+ clearance [138]. This effect was caused by NNMT induction resulting from activation of the SQSTM1-NF-kB signaling pathway.

Inflammation, a crucial factor in HF development, may affect autophagy, and vice versa [139]. Several studies indicate that the interaction between autophagy and inflammation plays an important role in HF progression [140]. Additionally, in this context, autophagy has dual roles, as it can either promote or inhibit NLRP3 activation [141,142]. Solute carrier family 26 member 4 (SLC26A4) could act as a link between inflammation, autophagy, and cardiac hypertrophy [143]. It was found that SLC26A4 activated autophagy and the NLRP3 inflammasome in a phenylephrine-induced cardiomyocyte hypertrophy in vitro model [142]. Furthermore, SLC26A4 activated the NLRP3 inflammasome in vivo in TAC rats. Therefore, targeting SLC26A4 expression to modulate autophagy and NLRP3 inflammasome activation may be a promising therapeutic strategy for cardiac hypertrophy and HF. Another study demonstrated that mtDNA escaping autophagy caused toll-like receptor 9 (TLR9)-mediated inflammatory responses in cardiomyocytes, leading to myocarditis and dilated cardiomyopathy [144]. Thus, impaired mitophagy could be a key mechanism driving chronic inflammation in failing hearts.

Several substances, including those used in traditional Chinese medicine, were tested as potential therapies targeting autophagy in HF, yielding promising results (reviewed in [22]). However, ClinicalTrials.gov lists only two clinical trials related to autophagy/mitophagy in HF, with one completed and the other in the recruiting phase, but neither directly aims to target autophagy in HF (https://clinicaltrials.gov/search?cond=heart%20failure&term=Mitophagy, accessed on 21 August 2025).

In summary, many studies indicate that effective autophagy has a beneficial effect in HF and its early stages, beginning with cardiac hypertrophy. However, excessive autophagy may be detrimental to heart health. It is crucial to understand that autophagy levels—normal, insufficient, or excessive—can vary depending on the stage of HF, particularly during the acute decompensated phase, when high autophagy activity might damage normal cellular components. Therefore, whether autophagy acts as a friend or foe in HF depends on the cellular context, much like in other syndromes.

## 5. Interplay Between Mitochondrial Quality Control, Sirtuins, and Autophagy in Heart Failure

The common factor among mtQC, SIRTs, autophagy, and HF is oxidative stress. It usually results from impaired mitochondria caused by faulty mtQC and is linked to excessive production of RONS, which damage cellular components that are targeted for autophagy. Sirtuins, due to their antioxidant properties, can reduce oxidative stress. However, SIRTs, especially SIRTs3-5 located in the mitochondrial matrix, regulate mitochondrial health independently of oxidative stress. Additionally, SIRTs play a role in autophagy regulation. Mitochondrial dysfunction, which can be influenced by SIRTs and autophagy, leads to an energy deficit in the heart, contributing to HF and its precursor syndromes.

As mentioned, impaired autophagy in mice caused by knocking out ATG3, a key initiator of autophagy, led to NAD+ deficiency in hearts due to increased NAD+ clearance [116]. Although the status of sirtuins was not examined in that study, it can be speculated that they might be impaired due to the lack of their cofactor, highlighting the importance of sirtuin–autophagy interaction for cardiac health.

Acute cardiovascular injury, such as that seen in myocardial infarction, can lead to acute HF [145]. An increased expression of SIRT7 was observed in mice in response to injury caused by myocardial infarction, especially at the active wound healing site [146]. SIRT7-deficient mice showed increased susceptibility to cardiac rupture after myocardial infarction and experienced delayed wound healing after skin injury. They also demonstrated less fibrosis, reduced fibroblast differentiation, and fewer inflammatory cells in the infarct’s border zone. In laboratory experiments, cardiac fibroblasts lacking SIRT7 expressed lower levels of fibrosis-related genes and had decreased transforming growth factor receptor I (TGFBR1) protein. The absence of SIRT7 induced autophagy in cardiac fibroblasts, and inhibiting autophagy prevented the reduction in TGFBR1. This indicates that SIRT7 may play a vital role in tissue repair by interacting with autophagy and TGFBR1.

As cited in earlier sections, myocardial IRI is directly connected to HF [147]. Briefly, Ca^2+^ overload, peroxidation, and inflammation are key factors in the development of myocardial IRI. The interaction among these factors leads to the activation of cardiomyocyte cell death pathways, including autophagy [147]. Therefore, targeting autophagy and other forms of programmed cell death may be an effective strategy to reduce IRI and, consequently, HF. Chloramphenicol succinate, an autophagy stimulant, administered to pigs that experienced coronary artery occlusion and reperfusion, resulted in a significant decrease in infarct size and increased expression of LC3 and another autophagy-related protein, BCN1, which also plays a role in regulating apoptosis [148,149]. Autophagy in cardiomyocytes is linked to SIRT1 through various pathways, including those involving ATG, LC3, and FOXOs (reviewed in [150,151]). Recent research has confirmed that SIRT1-mediated autophagy plays unique roles at various stages of myocardial I/R injury (reviewed in [150]). Therefore, targeting the mechanism of SIRT1-mediated autophagy with adjustments for a specific phase of IRI may be a therapeutic strategy for preventing HF development.

Hypoxic stress directly contributes to HF by causing myocardial damage and other effects related to HF, involving several signaling pathways, including HIF- and Na^+^/K^+^ ATPase [152]. Heart tissue samples from patients with cyanotic congenital heart disease showed increased autophagy, apoptosis, and higher SIRT1 levels compared to the noncyanotic control samples [151]. SIRT1 promoted autophagic flux and reduced apoptosis in hypoxic H9C2 cells. Additionally, SIRT1 stimulated AMPK, while the AMPK inhibitor Compound C blocked the activation of autophagy by SIRT1. SIRT1 protected hypoxic cardiomyocytes from apoptosis, involving inositol-requiring kinase enzyme 1α (IRE1α). The SIRT1 activator SRT1720 enhanced AMPK activity, suppressed IRE1α, increased autophagy, and decreased apoptosis in the heart tissues of normoxic mice compared to the hypoxic control group. Hypoxic mice treated with the SIRT1 inhibitor EX-527 showed opposite effects. Therefore, SIRT1 may promote autophagy through AMPK activation and reduce hypoxia-induced apoptosis via the IRE1α pathway, helping to protect cardiomyocytes from hypoxic stress and prevent HF development. This experiment exemplifies the regulation of cardiovascular autophagy by sirtuins. Initially identified for SIRT1, sirtuins can directly trigger autophagy by deacetylating autophagy-related genes or enhance autophagic flux by increasing the expression of genes that regulate autophagy [153].

Mitochondria might be connected to autophagy in HF through interactions with lysosomes, including regulating mitochondrial fission, influencing lysosomal transport, and mediating calcium signaling (reviewed in [72]).

Besides dysfunctional mtQC, protein quality control may also contribute to HF. It was demonstrated that impaired autophagic flux decreased the availability of NAD+, a cofactor of sirtuins, in cardiomyocytes [138]. NAD+ deficiency caused by impaired autophagy results from the induction of nicotinamide N-methyltransferase (NNMT), which methylates the NAD+ precursor nicotinamide (NAM) to produce N-methyl-nicotinamide (MeNAM). Administering nicotinamide mononucleotide (NMN) or inhibiting NNMT activity in autophagy-deficient hearts and cardiomyocytes restores NAD+ levels and enhances cardiac and mitochondrial function. Mechanistically, autophagic inhibition leads to the accumulation of SQSTM1, which activates NF-κB signaling and increases NNMT transcription. Therefore, this work suggests a novel mechanism showing how autophagic flux supports mitochondrial and cardiac health by mediating SQSTM1-NF-κB-NNMT signaling and regulating cellular NAD+ levels, resulting in changes in SIRTs activity.

Calprotectin, a heterodimer of S100A8/S100A9, is increasingly seen as a proinflammatory mediator in cardiovascular diseases, including heart failure (reviewed in [154]). Its involvement in HF development intersects with RONS production and sirtuin signaling, creating a complex network of metabolic and inflammatory disturbances. Released by activated neutrophils and monocytes, calprotectin acts as a damage-associated molecular pattern (DAMP), binding to TLR4 and RAGE receptors on cardiomyocytes and endothelial cells. This activates NF-κB signaling, resulting in the release of proinflammatory cytokines that promote cardiac remodeling and fibrosis. Calciprotein-induced inflammation worsens mitochondrial dysfunction in HF by increasing oxidative stress through immune cell activation and RONS production, forming a vicious cycle. Chronic inflammation and oxidative stress deplete NAD+ levels, impairing sirtuin activity. Calprotectin may indirectly hinder sirtuin function by sustaining low-grade inflammation and increasing ROS levels, which overwhelm sirtuin antioxidant defenses and disrupt mitochondrial homeostasis, leading to energy production problems.

In summary, the interaction between mtQC, autophagy, and sirtuins may contribute to HF development through various pathways. Dysfunctional mitochondria could be a key element in this process, as they are associated with increased oxidative stress, which can trigger autophagy to remove damaged parts of cardiocytes and activate SIRTs to reduce stress. Many compounds can influence this process, but mainly, they are canonical proteins of mtQC, autophagy, and SIRTs. Calprotectin plays a significant role in HF pathogenesis. Further research is necessary to clarify the specific interactions involved, which could aid in developing targeted therapeutic strategies based on this interaction.

## 6. Conclusions and Perspectives

Heart failure is the final stage of many cardiovascular diseases. Its development shares some aspects with these diseases but also has features that set it apart from other heart syndromes. A common factor for most, if not all, of these aspects is oxidative stress, which is mainly linked to dysfunctional mitochondria. These mitochondria may be characterized by impaired mitophagy, reflecting a compromised autophagic response of cardiomyocytes. Sirtuins are essential regulators of cellular and organism-wide responses to oxidative stress. As a result, oxidative stress, mitochondria, autophagy, and sirtuins may interact in the development of heart failure.

Currently, the literature broadly recognizes the protective role of autophagy in HF [155]. Autophagy removes damaged mitochondria that can increase oxidative stress through a vicious cycle, thereby reducing oxidative stress. This reduction is also aided by the antioxidative actions of SIRTs. Additionally, autophagy may lower the inflammatory response by degrading proinflammatory components, regulating inflammasomes, clearing pathogens and damage-associated molecular patterns, and modulating cytokine secretion [156]. Because of the important role of SIRTs in autophagy regulation, they might be involved in all these functions, providing a key element for HF mitigation through autophagy. Oxidative stress and inflammatory responses are critical aspects of ventricular remodeling, serving as hallmarks of HF. Conversely, excessive autophagy can accelerate this process, and further research should explore the involvement of SIRTs in this aspect of autophagy in HF development. Cardiomyocyte apoptosis, common in HF, may be driven by excessive autophagy and dysfunctional mitochondria, but SIRT3 controls mitochondrial outer membrane permeabilization, a vital step in the mitochondrial pathway of apoptosis [157]. Furthermore, SIRT1 was reported to have an anti-apoptotic role in hypoxic stress, which is characteristic of a significant portion of HF cases [158]. Therefore, further research into the mutual interaction and regulation of mitochondria, autophagy, and SIRTs is needed to develop potential therapeutic strategies for targeting HF.

The evolving nature of autophagy may impact HF and its precursors differently depending on the HF stage. Another factor to consider is the classification of HF into HFpEF and HFrEF. All these elements should be considered when planning HF treatment tailored to the molecular characteristics of the disease.

Although the involvement of mitochondrial dysfunctions in HF pathogenesis is documented, the relationship between RONS-induced mtDNA damage and its repair remains unclear based on current results, as the failing heart may compensate for oxidative stress effects by mobilizing DNA repair proteins to mitochondria [60]. Therefore, more research is necessary on DNA damage and repair in cardiomyocytes.

It is not surprising that sirtuins may interact with mtQC and autophagy, and this was not the goal of this work to demonstrate such an interaction in HF. Instead, we aimed to show the mechanism behind that interaction based on the roles of these three elements in HF pathogenesis.

As mentioned in the previous section, targeting SIRT1-related autophagy specifically during the phases of IRI might serve as a preventive and therapeutic strategy in HF. To achieve this, small-molecule drugs and miRNA regulators can be developed. Resveratrol, sevoflurane, quercetin, and melatonin could be considered during the ischemic stage, while coptisine, curcumin, berberine, and miRNA agomirs or antagomirs involved in regulating the SIRT1-autophagy axis may provide cardioprotective effects during reperfusion (reviewed in [150]).

Although our work involves effects and phenomena integrated into many normal and pathological conditions, it is currently difficult to identify which specific aspects of this integration contribute to HF. In fact, it is unlikely that there are features unique to HF that extend beyond the normal integrating circuit. For example, oxidative stress is bidirectionally linked to mtQC, produces RONS that damage cellular structures, and triggers autophagy. The level of this damage depends on the capacity of antioxidant defenses, which is also determined by the antioxidant activity of sirtuins.

In summary, mtQC, autophagy, and SIRTs may have significant potential in developing HF and its precursors. However, more research on their underlying molecular mechanisms is necessary to turn these mostly observational links into effective treatments and prevention methods for heart failure. The important role of sirtuins in HF development is suggested by several reports, but as they are relatively recent discoveries, additional studies are needed to fully understand their properties. Still, their role in controlling oxidative stress, mitochondria, and autophagy is well established, justifying their consideration in the context of heart failure pathogenesis and therapy.

## Figures and Tables

**Figure 1 ijms-26-09826-f001:**
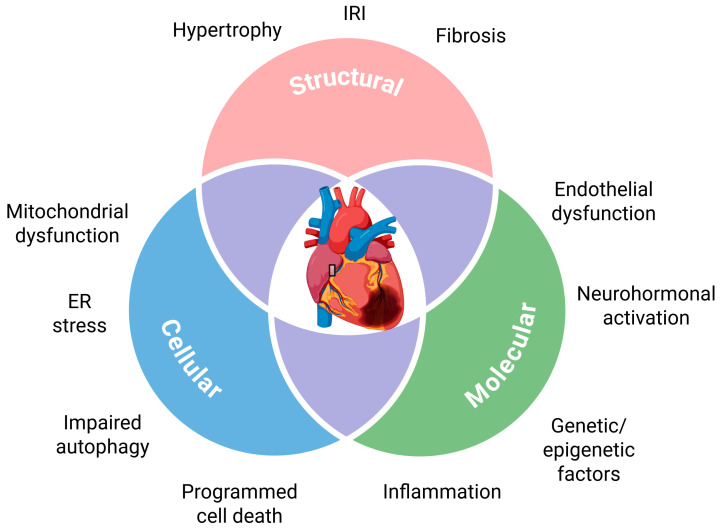
Precursors of heart failure. These precursors partly overlap, as shown in this diagram. ER stands for endoplasmic reticulum; IRI stands for ischemia–reperfusion injury. Created in https://BioRender.com.

**Figure 2 ijms-26-09826-f002:**
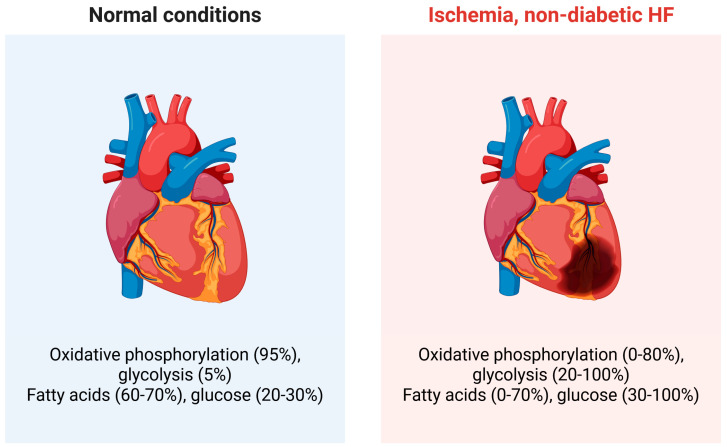
Energy in the normal heart is mainly produced through oxidative phosphorylation in mitochondria, with a minor role for glycolysis in the cytosol. However, in a diseased heart, such as an ischemic or failing heart, these proportions change as shown. These changes can vary widely depending on the specific disease and its clinical presentation. Significant differences exist in substrate use between diabetic and non-diabetic heart failure (HF), with the former relying more on fatty acid oxidation due to insulin resistance and impaired glucose uptake. Although only fatty acids and glucose are discussed here, ketone bodies, lactate, and other molecules can also serve as energy substrates. Created in https://BioRender.com.

**Figure 3 ijms-26-09826-f003:**
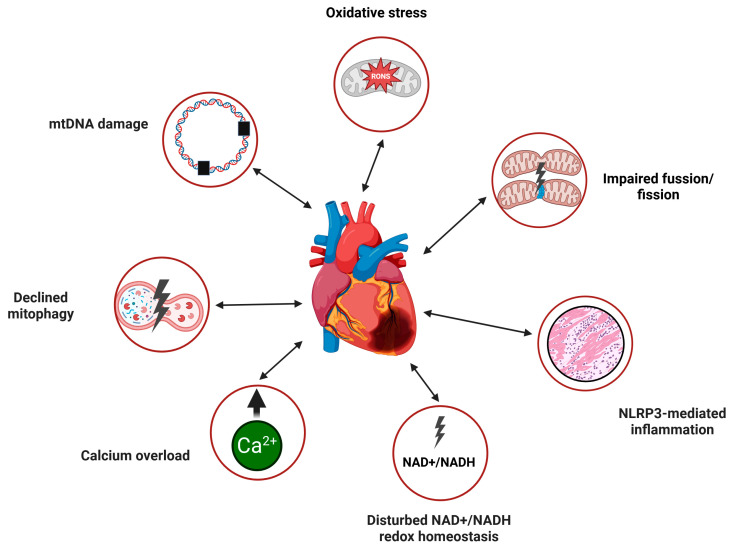
Mitochondrial dysfunction in heart failure. Two-headed arrows depict the mutual dependence between these dysfunctions and HF, indicating that, currently, only an association can be identified without a clear causal relationship. mtDNA, mitochondrial DNA; NLRP3, NACHT, LRR, and PYD domains-containing protein 3 (NLRP3); NAD+, nicotinamide adenine dinucleotide oxidized; NADH, nicotinamide adenine dinucleotide reduced. Created in https://BioRender.com.

**Figure 4 ijms-26-09826-f004:**
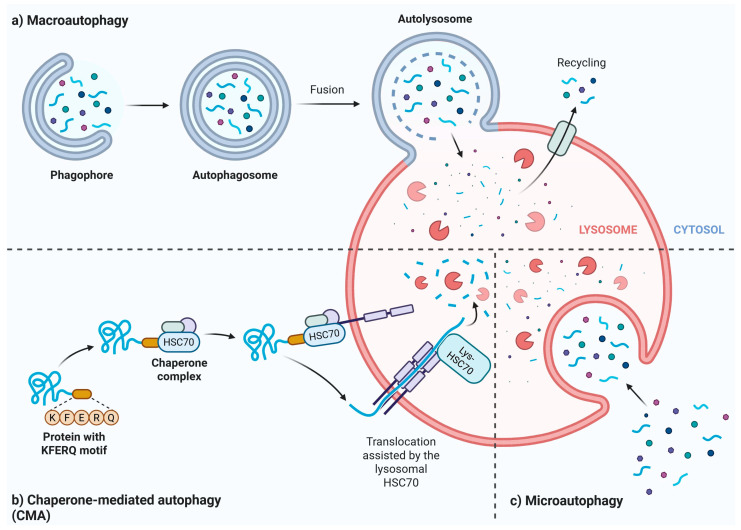
Outline of autophagy pathways. Macroautophagy begins with the formation of the isolation membrane (phagophore), which expands to form an autophagosome that encloses the cargo (a). In macroautophagy, the autophagosome fuses with a lysosome to form an autolysosome, where lysosomal enzymes digest the cargo. In chaperone-mediated autophagy, the cargo contains a specific motif, such as the lysine-phenylalanine-glutamic acid-arginine-glutamine sequence, recognized by a lysosomal membrane protein with the help of a chaperone, exemplified here by heat shock cognate 71 kDa protein (HSC70) (b). In microautophagy, the cargo is directly engulfed by a lysosome and broken down (c). Created in https://BioRender.com.

**Table 1 ijms-26-09826-t001:** Some heart failure-related effects of sirtuins.

Sirtuin	Subject	Effect	Mediators/Mechanism	Reference
SIRT1 ^a^	HFpEF patients	Reduced systolic blood pressure, increased EF	Increased levels of SIRT1 and NAD+ in caloric restriction and exercise program; increased antioxidant capacity	[85,86]
	Mouse model of HFpEF	Protection against harmful effects related to heart remodeling	Regulation of lipid metabolism and inflammation	[87]
SIRT2	*SIRT2^−/−^* mice and mice lacking SIRT1 in cardiomyocytes	Improved cardiac function after I/R and PO, reduced cardiac hypertrophy	Maladaptive effects in response to stress, decrease in antioxidant protection due to NRF2 inhibition	[89]
	*SIRT2^−/−^* mice	Cardiac hypertrophy and fibrosis, reduced EF and fractional shortening	SIRT2-mediated AMPK activity through deacetylating LKB1	[90]
	Mice with overexpressed *SIRT2*	Protection against hypertrophy and fibrosis caused by age and ANGII		
	Patients after acute myocardial infarction	Positive correlation between SIRT2 and HF		[91]
	*SIRT2^−/−^* mice	Heart hypertrophy, remodeling, fibrosis, and age-related dysfunction	SIRT2 interacted with and deacetylated the NFATc2 transcription factor	[92]
SIRT3	Mouse model of HFpEF	Fibrosis	hyperacetylation of mitochondrial proteins, resulting in enhanced production of interleukins IL1B and IL18 and increased assembly of NLRP3	[93]
	Mouse model of HFpEF	Impairment in mitochondrial fatty acid oxidation	Hyperacetylation of key enzymes of fatty acid oxidation, *SIRT3* downregulation, NAD+ deficiency	[94]
	Mouse model of HFpEF	Improved glucose uptake and metabolism	Activation of the skeletal muscle SIRT3-5′-AMPK pathway	[95]
	Mouse model of HFpEF with SIRT3 deficiency	Pulmonary vascular remodeling	Increased secretion of lysyl oxidase homolog 2 and β2-microglobulin	[95]
	HFpEF patients	Heart remodeling	Reduction in indole-3-propionic acid, activating NNMT-SIRT3 axis	[96]
SIRT4	Mice with ANGII infusion	Progression from compensated to decompensated cardiac hypertrophy	*SIRT4* overexpression	[99]
	Mice with heart-specific *SIRT4* overexpression	sped up heart failure development in response to pressure overload	Mitochondrial RONS-mediated increase in profibrotic transcriptional signaling	[100]
SIRT5	Mice with *SIRT5* overexpression	Protection against TAC consequences	Suppression of metabolic switch from fatty acid oxidation to glycolysis, immune activation, and fibrotic signaling pathways	[101]
	Mouse cardiomyocytes	Reduction in the cell viability, and an increase in the number of apoptotic cells and the caspase 3/7 activity	Direct interaction between B2CL1 and SIRT5	[102]
	mouse model of cardiac IR injury	reduced mitochondrial damage and alleviated cardiac injury	Increasing SIRT5 levels reduced mitochondrial damage and alleviated cardiac injury through interaction with ANT2, inhibiting its lactylation and enhancing its interaction with VDAC1	[103]
SIRT6	Mouse model of HFpEF	HF mitigation	restoring endothelial SIRT6 function and it was underlined by the deacetylation of histone H3K9 around the PPARG promoter	[104]
	Diabetic HF patients	Decreased level of SIRT6		[105]
	Diabetic mouse model of HFpEF	Improvements in diastolic dysfunction and decreased cardiac lipid buildup	Suppression of endothelial *PPARγ* expression via SIRT6-dependent deacetylation of histone H3 near the *PPARγ* gene promoter.	[105]
	TAC mice	Mitigated TAC-induced heart dysfunction and decreased cardiac inflammation, resulting in reduced cardiac fibrosis and smaller infarcts	Overexpression of SIRT6 elevated TERT and TRF1 levels	[106]
SIRT7	TAC mice	Increase in heart weight relative to tibial length, and they demonstrated a reduced cardiac contractile function	Interaction between SIRT7 and the transcription factor GATA-4 was identified, and GATA4 knockdown lessened the severity of phenylephrine-induced cardiac hypertrophy. SIRT7 deacetylated GATA4 in cardiomyocytes, influencing its transcriptional activity.	[109]

^a^ All abbreviations are defined in the main text.

## Data Availability

No new data were created or analyzed in this study.

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
