# Peer review of "Oxidative Stress, Mitochondrial Quality Control, Autophagy, and Sirtuins in Heart Failure"

_ijms, 2025, doi:10.3390/ijms26199826_

Round 1
Reviewer 1 Report
Comments and Suggestions for Authors
Krekora et al.
Heart failure (HF) is a growing health concern, particularly in aging populations, yet its underlying mechanisms remain poorly understood and effective therapies are lacking. Mitochondria, autophagy, and sirtuins form a critical network in HF development, with oxidative stress serving as a central link. Dysfunctional mitochondria may drive disease progression by amplifying oxidative stress and altering autophagy and sirtuin activity, both of which can have protective or harmful effects. This paper reviews the complex interplay among these pathways.
The paper is detailed and thorough; overall well written. Here are some comments.
- In page 5, “…increased NTN activation in the mouse failing heart under oxidative stress was also observed [40]. This effect might disconnect NADPH from ATP production pathways and hinder energy metabolism in the HF heart during oxidative stress.” NTN should be NNT. Also, it would be helpful to explain how this reflects a disconnect between NADPH and ATP.
- In page 9, it says “The liver kinase B1 (LKB1), a key kinase upstream of AMPK, was identified as a direct target of SIRT2. Consequently, SIRT2 promotes AMPK activation by deacetylating LKB1.” This is wrong; it is SIRT1, not SIRT2 (Ruderman 2010).
- If overexpression of SIRT4 increases HF, wouldn’t NAD supplementation also have a negative effect on HF?
Reviewer 2 Report
Comments and Suggestions for Authors
The review covers important interconnected mechanisms in heart failure pathogenesis, but in my opinion, there are several critical improvements that are needed before publication.
- The manuscript lacks clarity on literature search strategy, the inclusion/exclusion criteria. This is particularly problematic for a review claiming to be comprehensive. I recommend that the authors specify their search methodology and criteria for study selection.
- The epidemiological data needs updating with more recent statistics. Also, the definition of HF subtypes (HFpEF, HFrEF, HFmrEF) requires more precise ejection fraction ranges; the authors can use the official guideline as reference.
- The dual role of autophagy needs clearer delineation of when it's protective versus harmful. Given the focus on oxidative stress and its interconnections with mitochondrial dysfunction and autophagy, I strongly recommend incorporating a dedicated paragraph discussing the S100A8/A9 heterodimer (calprotectin) and its role in heart failure. S100A8/A9 has emerged as a critical damage-associated molecular pattern (DAMP) that bridges inflammation and oxidative stress in heart failure pathophysiology. The heterodimer promotes NADPH oxidase activation, enhances mitochondrial ROS production, and creates a loop that exacerbates cardiac dysfunction. Consider adding this subsection, and the author can use if they consider helpful, this recent review: doi: 10.3389/fimmu.2025.1630410. This addition would elevate the manuscript from a traditional review to a more contemporary analysis that incorporates the latest mechanistic insights in heart failure research.
- Insufficient critical analysis in some fields. For example, contradictory findings regarding PGC-1α levels in heart failure (page 6) are mentioned but not adequately analyzed. The authors should provide a deeper analysis of why these discrepancies exist and their implications.
- While the title promises integration of oxidative stress, mtQC, autophagy, and sirtuins, the actual interconnections are superficially addressed. I would recommend integrating this concept better.
Numerous grammatical errors and awkward phrasing throughout so I suggest to review it.
Round 2
Reviewer 2 Report
Comments and Suggestions for Authors
The authors have adequately addressed the reviewers’ comments, and the manuscript has improved considerably.